# Methicillin-Resistant *Staphylococcus aureus* Membrane Vesicles Inhibit the Proliferation and Induce the Apoptosis of Epithelial Cells

**DOI:** 10.3390/pathogens11121429

**Published:** 2022-11-27

**Authors:** Xu Chen, Jingwei Zhang, Meng Yang, Guanhuan Du, Fuxiang Chen

**Affiliations:** 1Department of Laboratory Medicine, Shanghai Ninth People’s Hospital, Shanghai Jiao Tong University School of Medicine (SJTUSM), Shanghai 200011, China; 2Shanghai Key Laboratory of Orthopaedic Implants, Department of Orthopaedic Surgery, Shanghai Ninth People’s Hospital, Shanghai Jiao Tong University School of Medicine (SJTUSM), Shanghai 200011, China; 3Department of Oral Medicine, Shanghai Ninth People’s Hospital, Shanghai Jiao Tong University School of Medicine (SJTUSM), Shanghai 200011, China; 4Faculty of Medical Laboratory Science, Shanghai Jiao Tong University School of Medicine (SJTUSM), Shanghai 200025, China

**Keywords:** *Staphylococcus aureus*, MRSA, membrane vesicles, apoptosis, epithelial cells, SSTIs

## Abstract

*Staphylococcus aureus*, or methicillin-resistant *Staphylococcus aureus* (MRSA), is the predominant pathogen in skin and soft tissue infections (SSTIs), and MRSA membrane vesicles (MVs) play a pivotal role in bacterial pathogenesis and the modulation of the host immune response. We aimed to investigate the interaction between MRSA MVs and epithelial cells. In this study, MVs were isolated from an MRSA culture supernatant using the ELD method, comprising an electrophoretic technique used in combination with a 300 kDa cut-off dialysis bag. The proteomic analysis of the MRSA MVs via mass spectrometry showed that shared and distinct proteins exist in the MVs from clinical MRSA isolates with different genetic backgrounds, such as health-care-associated MRSA (HA-MRSA) and community-associated MRSA (CA-MRSA). These MRSA MVs were found to suppress the proliferation and increase the apoptosis of HaCaT cells. We conducted qPCR array, quantitative real-time PCR (qRT-PCR), and Western blotting (WB) analyses, and the results indicated that BCL2 antagonist/killer 1 (Bak1) may be involved in the apoptosis of HaCaT epithelial cells. Our findings suggest that MRSA MVs inhibit the proliferation and induce the apoptosis of epithelial cells.

## 1. Introduction

*Staphylococcus aureus* (*S. aureus*), or methicillin-resistant *S. aureus* (MRSA), is a ubiquitous Gram-positive pathogen in both hospital settings and community settings responsible for superficial and invasive infections [1,2,3]. It has been reported that the health-care-associated MRSA (HA-MRSA) clone ST239 and community-associated MRSA (CA-MRSA) clones ST59 and ST338 are prevalent in Asian countries, such as China [3,4,5]. The various MRSA strains, especially CA-MRSA, usually cause skin and soft tissue infections (SSTIs), and it is essential to identify the MRSA isolates from SSTIs at the molecular level by molecular typing to define HA-MRSA and CA-MRSA strains, since CA-MRSA clones co-exist with HA-MRSA clones in hospital settings subject to dynamic changes and hospital–community interactions [5,6,7].

Extracellular vesicles (EVs) are nano-sized spherical bilayered vesicles produced by living cells, including bacteria, which are referred to by different names, such as outer membrane vesicles (OMVs) in Gram-negative bacteria and membrane vesicles (MVs) in Gram-positive bacteria [8,9,10]. MVs vary in size (20–100 nm), morphology, composition, and biogenesis and contain diverse components, including proteins, nucleic acids, and lipids. Moreover, it has been demonstrated that MVs can be produced by *S. aureus* [8,11,12,13] into the extracellular milieu during bacterial growth, which may be associated with the development or progression of diseases [13]. *S. aureus* MVs play a pivotal role in antimicrobial drug resistance, pathogenesis, and the modulation of the host immune response [11,12,13]. Recently, Bitto et al. advanced our knowledge of the immunostimulatory roles of *S. aureus* MVs [14]. *S. aureus* MVs contain DNA, RNA, and peptidoglycan molecules that can activate innate immune receptors and induce autophagy [14]. *S. aureus* is a predominant pathogen in SSTIs [15], but the role of *S. aureus* MVs in SSTIs, including MVs from HA-MRSA and CA-MRSA strains, remains unclear. In the current study, we investigated the components of MRSA MVs and their effects on skin epithelial cells.

## 2. Methods

### 2.1. Isolates and Molecular Typing

Clinical isolates of MRSA were collected from clinical specimens and preserved in a nutrient broth containing 30% glycerol. The MRSA strains were identified using the VITEK-2 Compact system, and molecular typing methods, including *spa*-typing, Staphylococcal cassette chromosome *mec* (SCC*mec*) typing, and multilocus sequence typing (MLST), in addition to detection of the Panton-Valentine leucocidin (*pvl*) gene, were performed as previously described [16]. The DNA extraction was performed according to the simplified alkaline lysis method [16]. Briefly, clinical isolates were diluted in phosphate-buffered saline and alkaline buffer and then heated to 60 ℃ for 45 min. After adding Tris-HCL and centrifugation, the supernatant was collected for further experiments. The *Spa*-type was obtained via http://spaserver.ridom.de/ by comparing the sequence of the amplified product with the primers 1113f (5′-TAAAGACGATCCTTCGGTGAGC-3′) and 1514r (5′-CAGCAGTAGTGCCGTTTGCTT-3′). The PCR conditions for *spa*-typing can be found at http://spaserver.ridom.de/. The SCC*mec* typing was conducted via the detection of *ccr* and *mec* gene complexes, as previously described [16]. The sequence type (ST) was determined from the MLST profiles that index each unique combination of alleles, and the MLST scheme developed by Mark Enright can be accessed at https://pubmlst.org/organisms/staphylococcus-aureus [17]. The HA-MRSA was defined as strains containing SCC*mec* I, II, or III. The CA-MRSA was defined as strains containing SCC*mec* IV or V in this study.

### 2.2. Isolation and Analysis of MRSA MVs

Equal numbers of colonies of clinical MRSA isolates were cultured in RPMI-1640 medium, and the isolation of *S. aureus* MVs from the supernatant was conducted using an electrophoretic technique in combination with a 300 kDa cut-off dialysis bag (ELD), as previously described [18,19]. In detail, 5 mL of filtered culture supernatant was loaded into a 300 kDa dialysis bag (Spectrum Labs, Fort Worth, TX, USA). The dialysis bag was placed in a gel holder cassette under a current of 300 mA (Tanon, Shanghai, China). After 30 min, the electrophoretic buffer was replaced and the electrophoretic direction was changed. After 2 h, the purified MVs were collected in an electrophoretic buffer containing glycine (7.2 g/L, Sangon Biotech, Shanghai, China) and Tris (1.5 g/L, Sangon Biotech, Shanghai, China).

The identification of MRSA MVs was performed via transmission electron microscopy (TEM). A concentrated MV (5 μL) solution was added to 200-mesh carbon-coated grids (Shanghai Taosheng light and electricity technology company, Shanghai, China) for 1 min. Then, the grids were dried with filter paper. For negative staining, 5 μL of 2% uranyl acetate was dropped onto the grids. After 1 min, the excess negative staining solution was removed. The samples were viewed with a TEM at 120 kV with a magnification rate of 49,000 times.

A proteomics analysis of the MRSA MVs was performed using mass spectrometry (Sangon Biotech, Wuhan, China). The bacterial proteins were denatured and enzyme-digested, and the peptides were analyzed using a Q Exactive plus LC-MS system (Thermo Scientific, Waltham, MA, USA). Th epeptide sequences and protein identity were determined by matching proteins in databases (UniProt Staphylococcus aureus) with ProteinPilot (v4.5) using paragon.

### 2.3. Cell Culture, CCK-8 Assay, and Flow Cytometry

The human skin epithelial cell line HaCaT (Fu Heng Biology, Shanghai, China) was cultured in RPMI-1640 medium with 10% FBS. After treating the HaCaT cells with MVs and treating the HaCaT cells with the electrophoretic buffer as a control, the cell proliferation was assessed based on a CCK-8 assay (Dojindo, Kumamoto, Japan). In detail, after exposure for the indicated time, 10 μL of CCK-8 reagent was added, and the optical density was read at 450 nm on a microplate reader (Bio-Rad, Hercules, CA, USA) after incubation for 1.5 h. The cell apoptosis was analyzed via flow cytometry with the use of PI and FITC-Annexin V antibodies according to the manufacturer’s protocol (KeyGEN BioTECH, Nanjing, China).

### 2.4. qPCR Array and Quantitative Real-Time PCR (qRT-PCR)

A qPCR array was used to detect the expression of apoptosis genes. Briefly, the gene expression profiles were analyzed using an apoptosis qPCR array according to the manufacturer’s protocol (Wcgene Biotech, Shanghai, China). Biological duplicates were performed, and the *β-actin* and *GAPDH* genes were used as endogenous controls. The data were analyzed using Wcgene Biotech software. A qRT-PCR was also performed to verify the gene expression results from the array, as previously reported [20]. The total RNA was extracted using the TRIzol reagent (Invitrogen, San Diego, CA, USA), and real-time PCR amplification was carried out as a two-step reaction. First, the cDNA was synthesized using the GeneAmp^®^ PCR system 9700 (Applied Biosystems, Foster city, CA, USA) with the PrimeScript^TM^ RT Reagent Kit (TaKaRa, Shiga, Japan), and then the real-time PCR was performed in the StepOnePlus system (Applied Biosystems, Foster city, USA) with SYBR Premix Ex Taq II (TaKaRa, Beijing, China). The triplicates were prepared for analysis, and the cycling conditions were configured as follows: initial denaturation (30 s at 95 °C), followed by 40 cycles of denaturation (5 s at 95 °C) and annealing (30 s at 60 °C), then a final stage including 15 s at 95 °C, 1 min at 60 °C, and 15 s at 95 °C. The primer sequences of the target genes are listed in Table 1. *GAPDH* was used as the endogenous control for the normalization of gene expression, and the relative gene expression was determined using the 2 ^−ΔΔCt^ method [21].

### 2.5. Western Blotting

The HaCaT cells were treated for 24 h with an equivalent amount of MVs based on the MV protein content and then collected and lysed on ice in RIPA buffer containing a phosphatase inhibitor and phenylmethylsulfonyl fluoride (PMSF). The protein concentration was determined using a Pierce^TM^ BCA Protein Assay Kit (Thermo Scientific^TM^, 23227, Waltham, MA, USA), and the samples were then boiled at 100 °C for 10 min. Equal amounts of total protein were subjected to SDS polyacrylamide gel electrophoresis (SDS-PAGE) and then electrophoretically transferred onto a polyvinylidene difluoride (PVDF) membrane (Bio-Rad, Hercules, CA, USA). The membrane was blocked in 5% bovine serum albumin (BSA) in Tris-buffered saline/Tween 20 (TBST) for 1 h at room temperature and incubated with the primary antibody overnight at 4 °C (Bak, sc-517390, Santa Cruz Biotechnology, Santa Cruz, USA; Bag-3, sc-136467, Santa Cruz Biotechnology, Santa Cruz, USA; or β-actin, A1978, SIGMA, St. Louis, USA). Then, the membrane was incubated with HRP-conjugated secondary antibodies for 1 h at room temperature. The bands were visualized using a Tanon 5200 CE machine (Tanon, Shanghai, China).

### 2.6. Statistical Analysis

Student’s *t*-test or one-way ANOVA, performed using Prism 6, were used for the statistical analysis, and a comparison was considered significant if the *p* value was less than 0.05.

## 3. Results

First, we verified the presence of MVs from MRSA culture supernatants via TEM. Shown in Figure 1A is a TEM image of the spherical bilayered MVs isolated from MRSA culture supernatants using the ELD method. The morphology and size of these MVs were similar to those documented in other reports [12]. Clinical MRSA strains usually cause SSTIs in patients. We selected two pairs of clinical HA-MRSA and CA-MRSA isolates (isolates 251 and 320; isolates 432 and 662) as representatives for a further analysis of the MV proteins. The results of the molecular typing of clinical MRSA isolates are shown in Table 2 and indicate that isolates 251 and 662 are *pvl*-positive CA-MRSA strains and that isolates 320 and 432 are *pvl*-negative HA-MRSA strains.

Then, we isolated the MVs from these two pairs of clinical MRSA isolates. CA-MRSA strains are reportedly more virulent than HA-MRSA strains. However, it remains unclear whether they exhibit differences in their MVs. By performing the MS analysis, we found that the MVs of clinical *S. aureus* isolates (pair one) with different genetic backgrounds contained both shared and distinct proteins (Figure 1B and Appendix A). By using log2 (exo-320/exo-251) as the lateral axis and log2 (MeanSP) as the vertical axis, the data in Figure 1B suggest that MVs from isolate 320 (exo-320) have more proteins than MVs from isolate 251 (exo-251), and the detailed information for each dot can be found in Appendix A. The top enriched proteins in the MVs from isolate 320 (HA-MRSA, exo-320) were aldehyde/alcohol dehydrogenase, assimilatory nitrite reductase, isoleucine-tRNA ligase, glutamyl-tRNA amidotransferase subunit A, carbamoyl phosphate synthase large chain, and cell division protein (FtsA). These proteins are mostly involved in bacterial growth and drug resistance. It has been reported that CA-MRSA strains harbor more virulence factor genes than their HA-MRSA counterparts [3], and in the current study, we found that the MVs from isolate 251 (CA-MRSA, exo-251) carried more α-hemolysin, lipase-2, lukS_PV, and clumping factor B, as indicated in Figure 1B. These data suggest that MVs from clinical HA-MRSA and CA-MRSA strains harbor distinct components, although the presence of some common proteins was observed. An MS analysis of the MVs from two pairs of isolates was also performed (data shown in Appendix A), and the common enriched proteins in the MVs from the two HA-MRSA strains and the two CA-MRSA strains are shown in Figure 1C. For instance, chemotaxis inhibitory protein (Chp) was commonly enriched in the MVs from the two CA-MRSA isolates, whereas other proteins identified in these clinical isolates were found to be unique. These results suggest that MVs from clinical isolates may differ in their components, despite being derived from isolates of the same category.

MRSA is a major pathogen causing SSTIs [15], but the effect of MVs from MRSA on epithelial cells remains unknown. We explored the interactions between MRSA MVs and epithelial cells. Based on the CCK-8 assay results of HaCaT cells co-cultured with different doses of MVs, we found that a low dose of MVs (1 μg) had no effect on the proliferation of HaCaT cells and that high doses of MVs (2 and 5 μg) from both CA-MRSA and HA-MRSA isolates inhibit the proliferation of HaCaT cells (Figure 2A). Then, three other doses of MVs (2, 5 and 10 μg) were used for the apoptosis assay. By gating PI^+^ and Annexin V^+^ subsets in flow cytometry data, we found that the MVs produced by these MRSA isolates induce the apoptosis of HaCaT epithelial cells in a dose-dependent manner (Figure 2B,C). The apoptosis gene expression profiles were analyzed using a qPCR array (Appendix A), and the results indicated that the apoptosis genes BCL2 antagonist/killer 1 (*BAK1*) and BAG cochaperone 3 (*BAG3*) may be involved in the apoptosis of HaCaT epithelial cells (Figure 2D). The results of the Western blotting showed that the levels of Bag3 protein did not differ in the HaCaT cells treated by MVs compared with controls, and that the Bak protein levels increase in the HaCaT cells when treated with MVs (Figure 2E). Bitto et al. reported that pattern recognition receptors (PRRs), such as Toll-like receptor 2 (TLR2), play pivotal roles in the response to *S. aureus* MVs [14], and we found that the TLR2 expression increased greatly after exposure to *S. aureus* MVs for 3 h (Figure 2F). Interestingly, the expression of apoptosis genes, including *BAK1* and *BAG3*, also increased greatly as early as 3 h into the exposure to *S. aureus* MVs (Figure 2F). Collectively, these results indicate that *S. aureus* MVs can inhibit the proliferation and induce the apoptosis of epithelial cells.

## 4. Discussion

In the current study, we successfully isolated MVs from the supernatants of clinical MRSA isolates using an electrophoretic technique in combination with a 300 kDa cut-off dialysis bag [16]. This method, named ELD, is more convenient than other methods, such as ultracentrifugation. The ELD method was first designed for the isolation of EVs from plants [18,19], and we also extended its application to isolate MVs from MRSA isolates.

We then investigated the components of MRSA MVs and found significantly different amounts of proteins, such as virulence factors, in the MVs from clinical MRSA isolates. To further explore the proteomic data of the MRSA isolates, we selected four isolates, including the two shown in Figure 1B, for mass spectrometry. From the proteomic data obtained via mass spectrometry and shown in Appendix A, we found that the proteins in MRSA MVs consist of enzymes that are involved in bacterial growth, as well as virulence factors. We also found that the minor proteins from the MVs of HA-MRSA isolates are associated with antimicrobial drug resistance, such as ileS and lactamase-containing protein. Interestingly, the chemotaxis inhibitory protein has been reported to be a unique virulence factor favoring the dissemination of CA-MRSA [22,23], and the proteomic data in Appendix A and Figure 1C reveal that the MVs of CA-MRSA isolates contain significantly more Chp protein. Other materials, such as small RNAs (SsrA, RsaC, and RNAIII), have also been found in MRSA MVs, indicating the potential role of MVs as a modulator of host–pathogen interactions [24].

It has been reported that variation among *S. aureus* MV proteomes influences their cytotoxic effect on host cells [25], indicating that the differences in MVs from clinical *S. aureus* isolates may account for their observed cytotoxicity in host cells. However, in the present study, the MRSA MVs from two different isolates could induce apoptosis and inhibit HaCaT cell proliferation. In the present study, based on a qPCR array and qRT-PCR, we found that *S. aureus* MVs could efficiently induce the expression of apoptosis genes, such as *BAK1* and *BAG3*, as well as the expression of proteins such as Bak1 in HaCaT cells. The proteomic data for the MRSA MVs showed that enzymes and virulence factors such as α-hemolysin, lukS_PV, and clumping factor B are present in MVs. Perhaps the enzymes that are required for bacterial growth cause epithelial apoptosis. It has been shown that lukS_PV can induce apoptosis in acute myeloid leukemia cells via the C5a receptor [26], which suggests that some virulence factors found in membrane vesicles may induce the apoptosis of host cells. However, the detailed mechanisms underlying apoptosis induced by *BAK1* require clarification through further investigations. These results suggest that the overall effects of CA-MRSA and HA-MRSA MVs are similar and that some proteins common to both HA-MRSA and CA-MRSA need to be further explored. A further genomic analysis or specific gene mutagenesis study is necessary to precisely identify the proteins or factors in MVs of both HA-MRSA and CA-MRSA that are responsible for cytotoxicity.

It has been revealed that *S. aureus* MVs contain biologically active β-lactamase and can play a significant role in the development of antimicrobial drug resistance [11]. Moreover, *S. aureus* MVs can be used as a delivery system for bacterial virulence determinants and can modulate immune cellular functions [13]. Recently, Bitto et al. demonstrated that *S. aureus* MVs contain DNA, RNA, and peptidoglycan molecules that can activate PRRs and can be intracellularly degraded via autophagy in host epithelial cells [14]. We also found that MRSA MVs can induce TLR2 expression in HaCaT cells, which implies a critical role for TLR2 signaling in skin infections. In brief, these results are suggestive of a biological role for MRSA MVs during skin infections. Taken together, our data provide evidence for the presence of virulence factors in the MVs from MRSA isolates and provide novel insight into the interaction between MRSA membrane vesicles and skin epithelial cells.

## Figures and Tables

**Figure 1 pathogens-11-01429-f001:**
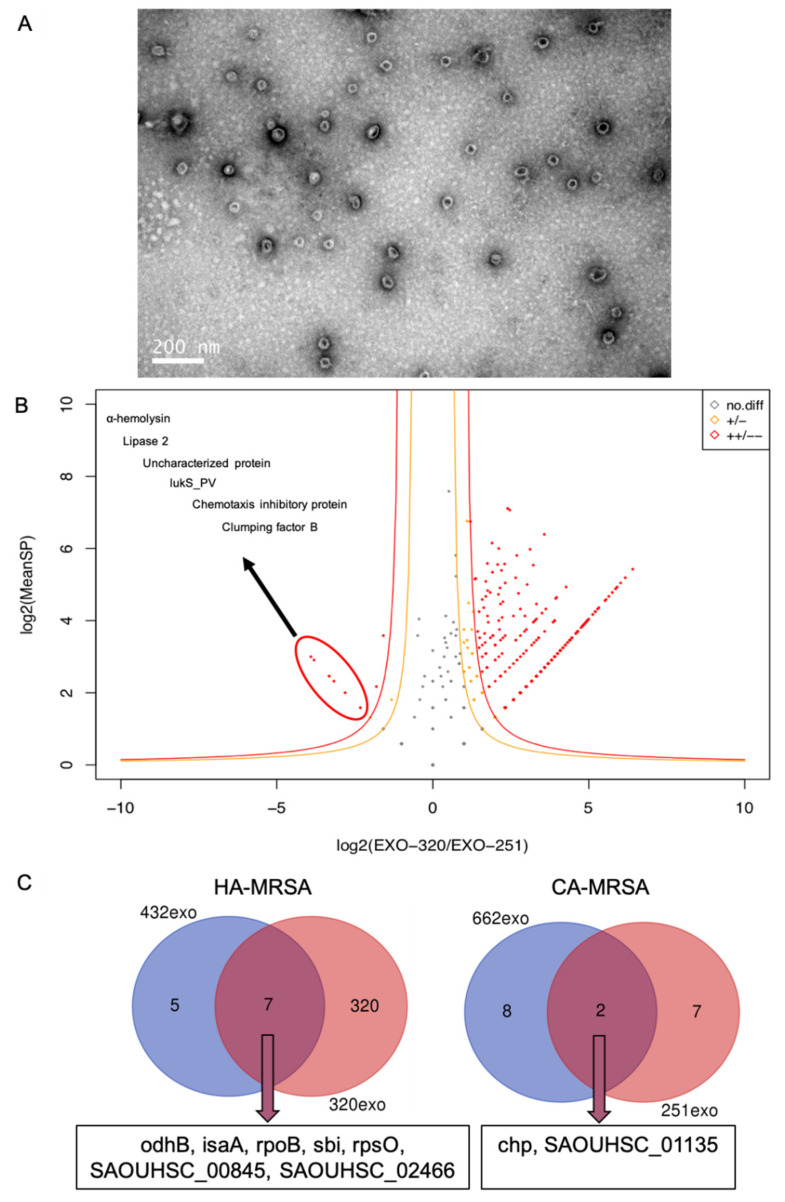
Production and proteomic analysis of MRSA membrane vesicles (MVs). (**A**) An image of MVs using TEM at 120 kV with a magnification of 49k. The MRSA was cultured in RPMI-1640 medium, and the MVs were isolated from the MRSA supernatant. (**B**) The mass spectrometric analysis of the MVs from different clinical MRSA isolates (isolate 251: *pvl*-positive t437-ST59-SCCmecIV CA-MRSA; isolate 320: *pvl*-negative t030-ST239-SCCmecIII HA-MRSA) and detailed information for each dot are shown in Appendix A. The gray dots indicate proteins that do not differ between these two MVs, the yellow dots indicate proteins that differ slightly between these two MVs, and the red dots indicate proteins that differ significantly between these two MVs. EXO-320 represents the MVs from isolate 320, and EXO-251 represents the MVs from isolate 251. (**C**) A Venn diagram of the commonly enriched proteins in MVs from HA-MRSA strains or CA-MRSA strains and their detailed information can be found in Appendix A.

**Figure 2 pathogens-11-01429-f002:**
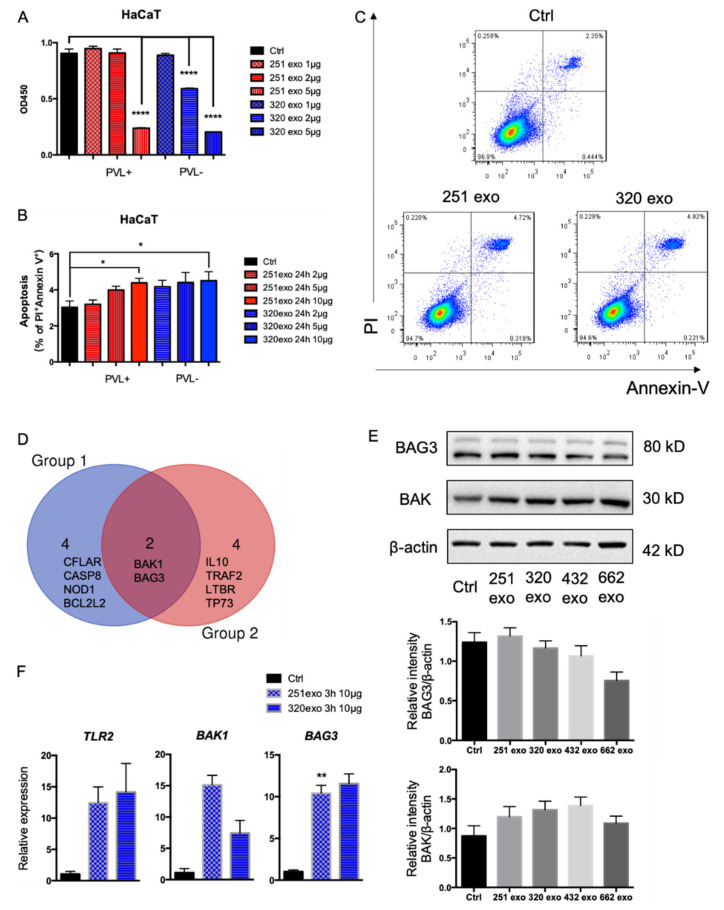
MRSA MVs inhibit the proliferation and induce the apoptosis of epithelial cells. (**A**) The proliferation of HaCaT cells assessed using a CCK-8 kit for the control or MVs as indicated. Three dose levels of MVs were used for the proliferation assays, 1, 2, and 5 μg. (**B**) Summary data for the HaCaT cell apoptosis for the control or MVs, as indicated. Three dose levels of MVs were used for the apoptosis assay, 2, 5, and 10 μg. (**C**) Representative flow cytometry data of the HaCaT cell apoptosis for the control or MVs (10 μg), as indicated. (**D**) Six leading apoptosis genes from two groups of MVs produced by *S. aureus* isolates (two isolates in each group; group1 included isolate 320exo and 432exo; group2 included 251exo and 662exo); the gene profiles were analyzed via qPCR array, and detailed information is shown in Appendix A. (**E**) Bag3 and Bak protein levels in HaCaT cells treated with MRSA MVs and controls analyzed via Western blotting. A densitometry analysis of target protein expression normalized to β-actin was performed using Image J. (**F**) The expression of *TLR2*, *BAK1*, and *BAG3* by qRT-PCR. Triplicates were assessed, and *GAPDH* was used as an endogenous control. * *p* < 0.05, ** *p* < 0.01, and **** *p* < 0.0001.

**Table 1 pathogens-11-01429-t001:** Primer sequences used for the qRT-PCR.

Gene	Primer	Sequence
*BAG3*	Forward	AGAGACGGTGTCAGGAAGGTTCAG
	Reverse	GTTGCTGGGCTGGAGTTCATAGAC
*BAK1*	Forward	AGAGATGGTCACCTTACCTCT
	Reverse	GGTCTGGAACTCTGAGTCATAG
*TLR2*	Forward	TGTCTTGTGACCGCAATGGTATCTG
	Reverse	TGCTAATGTAGGTGATCCTGTTGTTGG
*GAPDH*	Forward	CCTGCCAAATATGATGACAT
	Reverse	TCCACCACCCTGTTGCTGTA

**Table 2 pathogens-11-01429-t002:** Molecular characteristics of clinical MRSA isolates.

Isolate	Molecular Characteristics	Category	Note
Isolate 251	*pvl*-positive t437-ST59-SCC*mec*IV	CA-MRSA	Pair one
Isolate 320	*pvl*-negative t030-ST239-SCC*mec*III	HA-MRSA	
Isolate 432	*pvl*-negative t421-ST239-SCC*mec*III	HA-MRSA	Pair Two
Isolate 662	*pvl*-positive t437-ST338-SCC*mec*V	CA-MRSA	

## Data Availability

Not applicable.

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
