# Peer review of "Methicillin-Resistant Staphylococcus aureus Membrane Vesicles Inhibit the Proliferation and Induce the Apoptosis of Epithelial Cells"

_pathogens, 2022, doi:10.3390/pathogens11121429_

Round 1

Reviewer 1 Report

The paper by Chen et al. describes the effects of Staphylococcus aureus membrane vesicles on a human skin epithelial cell line.  In this study, a combination of proteomics and PCR techniques have identified changes tied to exposure to S. aureus membrane vesicles.  Overall, the data suggests  some profound effects.  Several things need to be addressed to make the paper more complete.

1.  Several methods are lacking important information.  An alkaline lysis method was used but there was no reference.  What PCR conditions were used for the spa-typing?  What is the source for the ELD? Carbon-coated grids came from where?  What magnification did you use for the TEM?  What is the source of the HaCaT cells?  How many replicates were done for the qPCR and qRT-PCR? What other controls were used for the PCRs besides GADPH? What PCR conditions and primers were used for the PCRs? Cite the paper for the 2-deltadeltaCCt method. What was the source of the PVDF membrane.

2.  Some of your figure legends have minimal information in the legend.  A figure is a stand alone entity that the reader should understand what is going on by looking at the figure with a complete legend.

3.  Figures 2A, 2B, and 2F should be normalized to protein level rather than volume.  The protein concentration was noted to be higher for the 320 strain compared to the 251 strain, so you do not have a true side by side comparison.  Figures 2A and B used different volumes.  Why?

4.  A densitometry analysis needs to be performed for Figure 2E to determine pixelation differences between lanes as normalized to the b-actin control.

5.  The discussion is too short and really does not discuss what proteins or other constituents of the MVs might be causing the changes that were seen.

6.  In the abstract, clarify that the PCR procedures and Western blotting was done to gauge eukaryotic cell differences.

7.  There were several grammar/wording issues.

8.  Acronyms need to be spelled out the first time they are used.

9. Western blotting has the W capitalized.

10.  Protein names start with the first letter capitalized.

11.  Two different styles of references were used and species names are italicized.

Author Response

Dear Reviewers,

Thank you very much for your professional and constructive comments and suggestions. We have revised the manuscript according to the comments and suggestions. And detailed responses to the comments and suggestions were in the attachment.

Reviewer 2 Report

In this manuscript, the authors have dissected the role of membrane vesicles of Methylene resistance Staphylococcus aureus on epithelial cells. This study is interesting because the authors found a key role of this membrane vesicles in inducing apoptosis of the host cells. However, the manuscript requires lot of editing in the way it is presented. I feel the authors can get help of any professional editors to rewrite the manuscript and resubmit. Instead of writing the results as a story, it felt like going through a bucket list of things they did. Even, they included materials and methods in the abstract.

Author Response

Dear Reviewer,

Thank you very much for your professional and constructive comments and suggestions. We have revised the manuscript according to the comments and suggestions. And we used the version with tracked changes to highlight the revisions. And detailed responses to the comments and suggestions were attached.

Round 2

Reviewer 1 Report

The authors have addressed my concerns.

Reviewer 2 Report

In this manuscript, the authors have dissected the role of membrane vesicles of Methylene resistance Staphylococcus aureus on epithelial cells. This study is interesting because the authors found a key role of this membrane vesicles in inducing apoptosis of the host cells. The authors have significantly improved the manuscript. Overall, the study is solid and is ready for publication. They have addressed all my queries.